# How Do We Address and Treat the Trauma of a 16-Year-Old Girl, Unaccompanied Minor, and Her Rape-Born Son? A Case Report

**DOI:** 10.3390/healthcare10102036

**Published:** 2022-10-15

**Authors:** Rahmeth Radjack, Luisa Molino, Anaïs Ogrizek, Elodie Gaelle Ngameni, Marie Rose Moro

**Affiliations:** 1Maisons des Adolescents-Maison de Solenn, Cochin Hospital, University of Paris, 75014 Paris, France; 2CESP—UVSQ, DevPsy, INSERM, Université Paris-Saclay, 94807 Villejuif, France; 3Consultation Trauma, Maisons des Adolescents—Maison de Solenn, Cochin Hospital, Référente Santé Mentale Médecins Sans Frontière (MSF), Chercheure Centre Babel, 75014 Paris, France; 4University of Sorbonne Paris Nord, 93430 Villetaneuse, France

**Keywords:** complex PTSD, case report, maternity, culture, migration, rape, psychotherapy, trauma

## Abstract

Background: The child psychiatry unit of the Cochin Hospital in Paris is specialized in a transcultural clinical approach and treatment of psychotraumatism. The clinical demands addressed to the service often combine several levels of vulnerability: recent migration, repeated and intentional traumas, isolation and breach in family bonds sometimes precarious living conditions. Mastering how to approach trauma content adapting to the person’s temporality while taking into account the individual, family and collective dimensions, is a key driver to the clinical intervention (of our approach). Objective and method: We describe a paradigmatic clinical situation articulating its multidimensional complexity: the case of Céline, a 16-year-old Mozambique teenager, unaccompanied minor (UM), who arrived in France three years ago with her 4-year-old child born out of rape. They are both cared for by Paris Child Welfare Bureau. The authors used the CARE guidelines for a rigorous approach to clinical case writing. Results and discussion: In the clinical discussion, we highlight the pertinence of transcultural abilities for the treatment of a complex PTSD (post-traumatic stress disorder). We describe the measures taken to adapt the clinical interview framework to the mother’s psychic temporality, while negotiating what can be said in attendance of the child. The idea of tranquility is primordial—whether she decides to tell or not tell the child. Removing the pressure to have to tell is an element of treatment. Conclusion: Working through a progressive narrative construction, the therapeutical process allowed for the restoration of multiple levels of continuity between times prior to the trauma and following it, as well as prior to migration and following it, to create a continuum from adolescence to adulthood. Restoring narrativity favors the process of becoming a mother and the one of negotiating this new identity. The therapeutic axes also focused on improving the well-being of the UM-mother and preventing the impacts of transgenerational trauma transmission to the child. For women with similar experiences, sharing their emotions and their stories with us makes their choice about telling their child legitimate and comfortable, regardless of the decision they make.

## 1. Introduction

Unaccompanied minors (Ums) are a vulnerable population at risk of post-traumatic stress symptoms, with intertwined post-traumatic, transcultural, family, educational, and legal issues [1,2,3,4]. They must grow up alone far from their family and the socio-cultural environment they initially come from. They have often experienced trauma before or on their migration journey, regardless of their reason for leaving.

In terms of clinical needs, their situation requires the provision of multidimensional care based on a support appropriate to adolescents, while taking family and transcultural dimensions into account.

Ums thus need specific mental health care, with cross-cultural competencies and a focus on the migration experience and complex trauma [4,5].

Dissociative and somatoform disorders, together with difficulties in trusting adults (including social workers and therapists), are regularly at the forefront of their symptomatologic experience.

Our department specializes in transcultural care and regularly receives patients referred either when their symptoms appear to be culturally coded, when the transcultural approach is necessary to create a bond difficult to establish otherwise [5], or at the request of other teams, when the first-line medical-psychological center (*Centres Medico-Psychologiques* (CMP): First-line community public service dedicated to primary psychological and psychiatric care) seems unable—for whatever reason—to disentangle the multidimensional complexity of the situation

In this article, we describe the therapeutic axes for the treatment of complex PTSD (Post-Traumatic Stress Disorder) in a young unaccompanied minor (UM) and her child conceived by rape, who also presented post-traumatic symptoms and developmental delay. Clinical challenges appeared on multiple levels: addressing the psychotrauma for each of them, protecting the young girl in her journey towards becoming a caregiver, being attentive to the disorders of the mother-child relationship, reducing the transgenerational risks of the trauma, accompanying the young woman in her development through adolescence, and taking into account the difficulties that the Ums must overcome to adapt to another country despite the insecurity of their immigration status.

The case described here Is paradigmatic of the complexity encountered in our clinic for unaccompanied minors in its combination of several levels of vulnerability: the experience of motherhood as an adolescent in exile, intentional trauma, social isolation, and social stigma. This case report deals with a topic that is not very common in the general mental health clinic, but not insignificant in the field of asylum and refugees. It shows how to improve the well-being of an Unaccompanied Minor (UM) mother with a child born out of rape, and how to try to prevent the impacts of transgenerational trauma transmission to his child. There is little research on this topic. The article is a description of 2 years of work from the therapeutic team. We describe the detailed steps of the psychotherapy led in a situation of trauma and cultural differences and what are the key concepts to know in order to reproduce this kind of treatment.

Young women represent a minority of uMs (about 25% in France, 9% in international migration [6]) for multiple reasons: boys might travel alone more easily, because they are less likely to be identified and included in the *Aide sociale à l’enfance’s* protection programs (public service for the protection of children and families in a situation of social and educational difficulties carried out by the State. In France, uMs may be placed in a group home, hotel or foster family), and because a non-negligible fraction of girls are caught up in human trafficking networks for the purpose of prostitution [7]. Thus, the prevalence of PTSD and other trauma-related symptoms is high among them due to events experienced premigration, on the journey, and/or post-migration [8].

Giving birth to a child conceived by rape is likely to have serious consequences on the wellbeing of any woman and on her relationship to her child, regardless of her age [9]. Hence, the main goals of a cross-cultural clinic are to provide care for the mother and to protect the child from the transmission of trauma [10,11,12] as well as to explore the individual and social meaning and representation of the traumatic events that disrupted their wellbeing. One of the major risks is that a mother distraught by intense suffering may be unresponsive or unable to invest emotionally in her newborn [13] and could thus reject the child [14,15].

We demonstrate here the axes of transcultural work using clinical levers that proved beneficial in this case and an approach proposed to cope with psychotrauma.

The therapist counter transference needs to be correctly grasped in this therapeutical approach since it can interfere in the way the interview can be led. For instance, the therapist may be guided by their own subjective reactions to the patient’s experience of trauma in order to find the tuning necessary for the construction of a narrative, on the condition that such reactions do not have a staggering impact.

The interviews are conducted according to a trauma-centered phenomenological psychotherapeutic approach (systemic and psychoanalytical) and to a transcultural framework aimed at facilitating a trusting relationship and the elaboration of a diagnostic and therapeutic narrative accessible to the patient.

To ensure the rigor of this report, the authors followed the CARE (Checklist of information to include when writing a case report) guidelines for the case presentation (see Appendix A). The patient gave us her consent for the publication of this case study. The CER-PARIS NORTH ethics committee (CEERB Paris North IRB00006477) approved this study.

Celine (this case study has been anonymized. The name as well as other elements linked to the patient’s identity have been changed) and her son have been seen in our department for 3 years, since she was 16 and he 3 years old, on a monthly basis. This case report is written chronologically to show the progression of the approach in psychotrauma and the adaptation of the clinical framework over time.

## 2. Case Presentation

At the time of our encounter, Celine was placed with the child welfare department due to her age and state of social isolation, as required by law for the protection of uMs in France. On the referral of a psychologist at that department, our child psychiatry team met with Celine, then 16 years old, of Mozambique origin, and her 3-year-old son Stephano. The referral request specified the need for “a space for Celine to talk, as well as support in navigating her status as a young mother with an unwanted child born out of an act of violence”.

The social workers said that Celine had been experiencing difficulty in investing emotionally in her child and had appeared to be in a state of traumatic shock for several months.

Celine and Stephano had been in France for a year at the time of our encounter. By the time they were identified and taken into an emergency shelter, they had spent several months wandering the streets. Celine was described as sad, withdrawn, inhibited, and sometimes even mute when she entered the child welfare system, while Stephano appeared agitated. She appeared to be a pleasant young woman. She had little contact with her son and avoided looking at him. In return, he did not make much contact with her.

Stephano was conceived by rape, and Celine was finding it difficult to establish a mother-son bond with him. The foster mother was providing most of his care and would go on to become an important attachment figure. This caregiver speaks Portuguese, as does Celine, and comes from a cultural group (not specified to ensure her anonymity) close to that of Celine and Stephano—both positive factors for Celine. Celine has a room inside the foster home. The foster mother also hosts another teenager and leaves with her husband. It is important to note that she had to go through the death of her own child in his early childhood. The foster mother is supervised by an educator from foster care and by the psychologist from the institution.

Celine subsequently described her reason for consulting: “it’s to get along with Stephano”. She used subtly chosen words in front of Stephano to signify that she wanted to care about him, or knew she should, but was unable to.

During her first year of therapy, she had no contact with Mozambique, either by telephone or internet. Her only friends were in France.

The interviews were conducted in the presence of either a Portuguese or Lingala interpreter. At the beginning, Celine’s mastery of French was limited; her Portuguese was fluent, and she also spoke Spanish, English, and Lingala.

At the first meeting, we explored her experience of arriving in foster care, and her current life context. Then, gradually, we cautiously began to explore her premigration experience, without targeting the traumatic event. We told her that she was not obliged to disclose unwanted details and that we would proceed at her pace. We encouraged her to warn us should she feel uncomfortable or emotionally overwhelmed at any time, explaining that if that occurred, we could continue another time and only if it made sense for her.

Before the interview, we stated clearly that patients are not under the obligation to discuss uncomfortable details. These statements have proved very useful in overcoming patients’ dread of the consultations due to fear of reactivating trauma or being psychologically distressed.

During this first session, we took the time to get to know Celine and explore her own childhood, who she was before leaving, as a child and teenager in Mozambique, the configuration of the family, her place among her siblings, and her current significant network. Celine is one of three siblings, and her parents had been separated for several years. When we asked her about her schooling and her future plans, she told us that she had been in high school but she had to stop “after what happened”.

At that point, Stephano, who then had primary enuresis, urinated in the interview room while playing with the nurse. This enuresis may have represented a somatic expression of anxiety or trauma. He was taken out of the room to be cared for with the help of the foster mother; a nurse played with him in another room while the consultation with Celine continued.

In Stephano’s absence, Celine revealed more elements of her post-traumatic symptoms: late night insomnia and post-traumatic flashbacks that have diminished in intensity over the course of therapy but were then preventing her from concentrating at school. She was having recurring nightmares and bruxism at night, resulting in intense toothaches during the day, but she did not dare disclose it to the foster mother. Celine described herself as withdrawn, wanting to isolate everything related to the trauma, out of shame and modesty. She wished to protect others around her and did not want to cause pain or worry. She self-soothed by listening to religious music. Combining the elements provided by the educator team who referred Celine to us, and our own clinical observations, we can describe her main symptoms as corresponding to a complex PTSD: vivid flashbacks, nightmares, difficulty controlling her emotions and elements of physical dissociation (emerging gradually as our therapeutical sessions go by), feeling very angry or distrustful towards the world, avoiding relationship, constant feelings of emptiness or hopelessness and feeling as if you are permanently damaged or worthless. She also suffers from distress without suicidal ideas.

She finally revealed that she had been raped by a policeman. After this event, her father, in a rage, had decided to file a complaint against the policeman. She had been taken to the hospital because of her injuries. It was subsequently discovered that she was pregnant. Her father had wanted an abortion, while her mother had encouraged her to continue the pregnancy. Celine’s reliving of her trauma concerned the rape but also her parents’ separation; it reactivated the feeling of emotional insecurity she experienced during family quarrels and disagreements.

Celine described threats to her family that resulted from the police complaint. The parents helped her run away while pregnant, entrusting her to a friend of her father to protect her. He was the only one to know that she had given birth. When we met her, she had not heard from her parents for several years.

Stephano was born at a hospital with the help of a doctor in this friend’s network just before she had left Mozambique. His name was chosen from the names listed by the doctor who had delivered him, after Celine told him that the baby would be named “nobody”. It took them 3 years to arrive in France.

Celine described her difficulties with Stephano in her own words, and how they had evolved over time: at first, she could not touch him, but now, at this first session, she was able to call him, and he would come to her; they could play together a little. She could not put into words what she then felt towards Stephano.

We asked her about the system of filiation in Mozambique to understand Stephano’s place in the line of filiation and his identity. In the region her parents come from, children’s education and affiliation may follow both the matrilineal and patrilineal lines.

Celine stated her willingness to come back after this first consultation, saying that she wanted to “get the images out of my head”. Throughout the consultations, she continued to elaborate on these events.

We invited Stephano to come back in the consultation room and together we told him that we had discussed the journey they took, talking about what he could understand and what would be useful for him to know.

While we had been talking to Celine alone, Stephano had built a house with building blocks. Hence, we talked about the concept of home. Using a short story, we discussed the process of re-building a home for people living far from their home country. Metaphors and traditional storytelling are used in transcultural consultations to help the child metabolize specific concepts, provide a sense of holding, and a closure to the consultation [16].

## 3. A Narrative Facilitating the Diagnostic Formulation for Celine and Stephano Simultaneously

At the following session, Celine appeared relieved, more open, and talked more spontaneously. She also reported finding Stephano calmer and that she was no longer having nightmares. It appeared as if she had been reassured by our ability to share her trauma [17], our welcoming her by showing that we could listen to her story, and our wanting to keep a protective link with her.

She told us that the flashbacks persisted but were modified: she was now seeing herself and her family as dead. At this stage of the therapy, she talked about her family, this time in front of Stephano. She had not seen them for 3 years. I informed her about the Tracing services provided by the International Red Cross (Tracing service and Restoring family link program provided by IRCR https://familylinks.icrc.org/, accessed on 15 January 2020), but she did not feel ready to inquire about their whereabouts.

The fear of receiving bad news, at a time when all personal energy must keep focused to move forward in France, often postpones the time to undertake such procedures for our UM patients. Similarly, the fear of revealing to family members the difficulties of their living conditions in France, or the wish to hide them out of respect for the family’s difficulties or sacrifices to finance the journey, plays an important role in the fading of family ties.

The last news Celine had had from her family was not reassuring; she had learned through a friend on social networks that her mother had been hospitalized due to a decompensation of her diabetes. Subsequently, she had no more contact with this friend.

During the consultation, we noticed that Celine looked away from Stephano. She noted that was finding it “burdensome” to take Stephano to school in the morning. In her country, she said, the family would have carried out this task, among others.

We decided to find a way to spend some time alone with Celine. Over a few consultations, we reorganized the setting as follows: a time of interview in a professional triad with Celine and in the presence of an interpreter, a space for Stephano to play or draw, assisted by a nurse on a children’s table set up next to her, and near the foster family, and then a time when he would leave the room with the foster mother. The choice of the moment for him to exit was relatively unstructured. We identified it by paying attention to verbal or nonverbal signals, when we sensed the prodromes of a traumatic or difficult story that Celine seemed not to want to reveal in front of Stephano, or as soon as he showed signs of fatigue. In fact, his attention was labile, going from one game to another agitatedly. Sometimes he played noisy games, perhaps to avoid hearing what was being said. His communication was discontinuous.

Celine said that she wanted to be considered a big sister until her 18th birthday. At that time, Stephano called her by her name and called the foster mother “*maman*” (mom). Nonetheless, according to Celine, Stephano knew who his real mom is. In her cultural group, all women who take good care of children are called “*maman*”.

At the third consultation, Celine discussed the content of her flashbacks, which were mainly about the childbirth or the separation from her own parents. “It wasn’t a normal delivery, it was forceps first, then a C-section”. She described thinking she was going to die, that a few hours later, white foam would come out of her mouth and she would convulse.

We interpreted this as the deadly dimension of the rape and a bodily reactivation of the traumatic trace. We also suspected that there had been obstetric complications, such as eclampsia or preeclampsia. Further, it appeared possible that Stephano showed signs of acute fetal distress.

Celine used positive terms to describe Stephano: curious, exploring a lot, calmer. Nonetheless, he appeared to present a speech delay and a behavioral disorder. We decided to conduct several psychological tests to determine the causes of her behavioral disorders (exploring affective and cognitive dimensions), underscore a delay in her learning achievements and skills: a cognitive development test (WPPSI-IV), psychomotor tests, the Bender test to assess visuo-motor functioning, R. Zazzo’s *Epreuve du bestiaire* (a test evaluating the affect, attitude, and social representations of preschool children), personality tests with the Rorschach (Rorschach, CAT and Sceno test game are psychological tests used to evaluate the thinking structure and emotional functioning) and CAT tests, and the Sceno test game. Stephano’s tests have been carried out by psychologists specialized in child mental health and trained to these tests.

We also performed a speech evaluation, using the ELAL from Avicenne, a tool specifically designed to be the first transcultural tool internationally available to evaluate the speech competence of multilingual children that includes and values their mother tongue [18]. The results highlighted a speech delay.

The cognitive development test (WPPSI IV) results were borderline low/weak.

The rest of the assessment showed Stephano’s confusion at several levels, which are representative of the effects of trauma on this 4-year-old child, including the confusion of generational reference points, as well as subject/object and temporal confusion.

During the assessments, he made enigmatic drawings of pipes and labyrinths, probably in connection with his migratory journey. The discourse around these drawings was disjointed and bore the traces of the multiple traumas affecting him since birth or even earlier.

The Rorschach test reactivated a high level of anxiety. His speech constantly evoked invasive traumatic reminiscences, although it was not clear to whom they belong. The so-called maternal boards evoked associations with death.

During the Sceno Test Game, Stephano reproduced games evoking traumatic scenes. He played a man (*un Monsieur*) named *Mean* who beat the children. Next, this same man became *pai* (word for daddy in Portuguese) who came to protect the children threatened by a car that crushed them one by one. Stephano identified with this car, and later in the game he became powerful and uncontrollable. Here, we found the confusion of the sense of values of complex PTSD and the difficulty of distinguishing malevolent from benevolent individuals.

## 4. Psychotherapeutic Interventions: Coming out of Isolation, Accepting Anger, and Finding Meaning in What Is Happening

After 5 interviews, a child psychiatric therapeutic space was set up specifically for Stephano. Regular clinical updates were carried out in between our sessions with Stephano’s child psychiatrist. In addition, he was able to participate in therapeutic mediation workshops, adaptive schooling (classes for children with special needs), and speech therapy.

At the same time, Celine began training to work in the field of early childhood care to, she said, “better understand my son”. This was the first time that she called him “my son” in a consultation, and she said it in French. Our clinical experience suggests that expressing certain words in one’s native language can be experienced as frightening: the trauma is embedded in the native language and using a second language may be experienced as protective [19].

Cecile’s progress in French was rapid thanks to her commitment to language classes in a community organization, but also to her ability to recreate links, to be less isolated, which we praised. Much later, she was able to engage in romantic relationships after overcoming a phase of mistrust towards others.

Separating the therapeutic spaces for Celine and Stephano seemed to allow Celine to express more intimate emotions and experiences, without fear of judgment. She told us that Stephano reminded her of the rape and that their foster mother was afraid to leave her alone with him for fear that Celine will not care for him. She allowed herself to say that *he is a pain, he screams, runs, and “behaves badly*” (because he is very agitated) on a daily basis. She withdrew, especially at the time of the year around Christmas, her own sister’s birthday. She associated it with the lack of family, but also with the traumatic reactivation of the rape that had occurred around this time of year. She described dissociative symptoms: “a part of me goes out (of my body)”. At this point, we decided to include the foster mother at our therapeutical sessions in order to all share our emotional experience together. We also organized a review meeting with all the different educators in attendance of the foster mother.

Celine gradually developed a co-mothering role; however, she did not yet trust her feeling or feel confident in this role. At the beginning of her life with the foster mother, she had watched Stephano outside playing from the window of their apartment and observed his behavior in an attempt to understand him better. Later on, the outbreak of COVID-19 and the lockdown that followed allowed for closer interaction, real encounters she felt ready for. They were able to make pancakes and watch movies together as she did not feel “forced” into this exceptional climate of confinement that destabilized the rhythm of daily life. Her sleeping improved over this time as well.

We continued further discussions about the differences in child-rearing habits *here* (in France) and *there* (home country) and tried to build some continuity between the two worlds; Celine began to sing lullabies from Mozambique to Stephano. In this phase, we were able to explore the subject of her own childhood and intrafamily relationships further. These discussions probably prepared her to reconnect with her family, as Celine started to proactively look for news and was able to give some soon after.

She also expressed strong feelings of anger. When we asked her to analyze them further, it emerged that they were mostly related to the experience of abandonment and to her parents’ separation. More specifically, this anger was related to conflicts in loyalty toward her parents: her father had had an extramarital relationship when Celine was younger. Celine was angry, she finally concluded, with her mother for letting her father marry the other woman. She also evokes an ambivalent feeling of anger related to the fact that her parents organized a plan to protect her, but this plan actually resulted in separating them from each other and her from her country (fleeing the country due to threats).

At this point, she still wanted to wait until she turns 18 to obtain the contact information of the Red Cross as she was not yet ready to see her parents. She feared her family’s reaction, mostly that her mother’s response might be melodramatic. She could picture her mother fainting, as she did when she saw an aunt after several years of separation.

We prompted her to describe the intra-family relationship in more detail.

Celine was the last of her siblings: “I was the princess at home. I was often sick”. She said she was close to her father until she was 8 years old, when her parents separated, and then she became close to her mother. After the separation, her father often called her mother to check on Celine because of her fragile health.

In transcultural consultations, we often find a singularity among our patients, that is, their sense of having a privileged or special place in the family. This element often allows us to identify an etiological theory, a possible re-interpretation charged with meaning of the events framed in cultural terms, beyond the medical reading that leads to complex PTSD.

From this perspective, we can say that some elements of Celine’s story indicate that the family decided to keep the singular child away from them to protect her. As we proceeded with our clinical investigation, Celine mentioned a witchcraft attack for which her stepmother was responsible. Specifically, we learned that Celine was involved in a car accident when she was 11 years old. Both her mother and Celine believe that her stepmother intentionally tried to run her over with a car, an event that created still more conflict between the parents. Her stepmother was preventing her father from seeing Celine’s mother. Celine had been living with her father for some time, but because of the disagreement with her stepmother, she had returned to her mother’s home, and that is when the rape occurred.

## 5. Follow-up and Outcomes

In the second year of child psychiatric follow-up, we observed Celine’s subjectivity becoming more structured, as she became physically and psychologically more autonomous and assertive. She came to her individual interviews by herself, and she accompanied Stephano to his child psychiatric interviews with our colleagues, although previously they had both been routinely accompanied by the foster mother. We observe a positive evolution regarding the complex PTSD symptoms; Céline can invest relationships with pairs and considers having a boyfriend. Flashbacks and nightmares have decreased, but they tend to reactivate in specific time of the year related to traumas. She tends to isolate herself only at festive seasons fore it reminds her the deprivation of her family, but this feeling of isolation progressively decreases as she tends to relate to other groups (foster mother, friends, internship). She has a better self-esteem and no longer has feelings of distress.

She is now comfortable being alone with Stephano. She considers that the foster mother overprotects him, she respects her role, but Celine has begun to assert her own nurturing skills: “my older sister entrusted me with her 5- and 6-year-olds”. Moreover, “when Stephano was a baby, I learned to change his diapers by myself”.

The therapeutic process has enabled her to rediscover an image of herself as a valuable subject with skills and competences that she can now acknowledge and trace back in her past. She is also self-aware of her limits, which she asks to work on: she appears to have some psychomotor slowness, and describes herself as passive, having been overprotected, and therefore not taught to take initiatives. Celine also distinguishes between the child-rearing techniques she knows from her country, and the way of taking care of children in France, and she blends the practices: “In Africa, we don’t play with children. Children play with children, that’s not my culture”.

In therapy, we value and reinforce her progress, although she is not always able to acknowledge it; she is working to overcome her traumatic freezing. She has begun to look for her mother through friends and her brother. She has learned that the policeman who raped her died in an accident. Both her brother and her father have gotten married, while her mother continues her business, but is sick. Celine feels guilty for not being able to see her. No one in the family talks about the rape, but this is a relief for Celine. A form of continuity between family bonds and friendships, between the dimensions of *before* and *after* the rape, seems now possible. According to her, back home, she would have been blamed for being raped: “Africans trivialize rape”, she told us.

Her professional and training projects benefit from this new dynamism as well. Celine appreciated her internships in a kindergarten with older children, which she preferred to the one in the nursery with babies, and she is considering continuing in this field for the future.

Stephano’s behavioral disorders are improving. In terms of language, his Portuguese is fading, but he speaks French and continues to see a speech therapist.

Celine is putting off until later her definition of her maternal role regarding Stephano. At 18, she says in Lingala, “I will see him normally, as my soI. Now, I can hug him. Before I didn’t feel anything, I was laughing defensively”. She can see herself and Stephano in five years: she imagines working in a school and having a good relationship with a boyfriend.

She foresees Stephano asking questions about his father in the future. She thinks she will answer that “*il n’y a pas de père*” (“there is no father”).

Celine has questions about the end of the therapy and wonders whether Stephano may be permanently traumatized by their life and migration journey.

## 6. Discussion

### 6.1. Appropriate Progression of the First Interview of Patients with Complex Psychotrauma

This interview is conducted applying a transcultural approach that uses an interview guide developed by combining our department’s clinical practices and international references [20,21,22,23,24,25]. This guide suggests 10 points to explore to enable the establishment of a relationship of trust in a transcultural situation (cf Table 1), promote a rich narrative, and make a start on cultural *metissage* (often useful for UMs in constructing their identity). By *metissage* we refer to an identity building processes that integrates feelings of belonging to multiple cultural univers. It may be common among children from mixed couples, couples from migrant background, or among children directly confronted to migration which tend to progressively integrate some of the cultural norms of the host country.

Facing the stress of adapting the hosting country’s culture, a harmonious cultural *metissage* is protective. It consists of keeping an ethnic affiliation to one’s culture of origin, while adopting new cultural affiliation from the hosting country, in a creative process of integrating values from each culture. This *metissage* process is very complex and multidimensional, including individual dynamics but also on a psychological, family, social point of view [26,27].

Showing our esteem for the patient’s representation could be the first step to correctly accommodate them therapeutically speaking. Reception must be conceptualized according to the patient’s cultural background: group welcoming can be foreseen if the patient comes from a country where problems are handled together, welcoming a husband alone before his family, agreeing for a whole community to rally. Taking the time to show how interested we are in their story, to gather information and listen to what they have to say, giving them a central role in the therapy in order to build together and learn from them. This will play a huge part in giving them back positive feedback that will help them to recover [24].

In situations of psychological trauma, we take the precautions needed to ensure that this narrative is not experienced as forcibly imposed, which could massively reactivate the trauma. The constant demand for their story of migration to which migrants are subjected along their trajectory at multiple institutional levels, for clinical and/or administrative reasons, proves to be a great challenge for many of our patients. In this kind of clinic, we take the time to specify that retelling it is not strictly necessary; it will happen only when the person is ready and if it is clinically relevant.

The Iim is to engage patients dynamically and authentically, by exploring their cultural identity and history to help them to escape a thought process frozen by trauma. Together we reconstruct the network they belong to and bring them into the consultation, even if they are far away, especially for UMs.

In circling around it we reach the heart of the trauma, similarly to peeling an artichoke to get to its heart. Being too direct creates a risk of forcing the narration and retraumatizing the patient.

By exploring the standard basis of meetings with adolescents (school, languages), the therapist shows interest in who they were before and still are. This procedure also makes it possible to mitigate the risk of “migration splitting”. Nathan (1986) described migration trauma as a sort of new psychic and sensory envelope to be reconstructed in migration [28]. This separation between before and after migration generates narratives where migrant patients tend to tell their stories starting from their arrival in the host country, obscuring earlier aspects unless they are actively asked about them.

For traumatic situations, several levels of resistance must often be overcome, and especially the role of the narrative must be made explicit. The story is gradually rewritten [29]. Some patients come in with a sort of magical thinking, imagining that they can forget the trauma if they don’t talk about it anymore, despite the massive functional discomfort associated with post-traumatic symptoms. We explain that it is difficult to forget and that, on the contrary, repression raises the risk of its subsequent reactivation at a moment of vulnerability or of exposure recalling the trauma; this reactivation is not only uncontrollable but can also be of even greater intensity than the initial trauma. What we are going to work on is “helping them to live with” the trauma, and to mitigate the consequences. We do not make them recount the story unless it is necessary, if there is a functional discomfort to try to transform.

By creating a safe place and providing culturally competent care, it is possible to focus on the narrativity of the trauma once more basic needs are addressed.

Concerning the cultural explanatory models of the problem, it may be possible to say that Celine’s narrative about the accident is consistent with the transcultural and anthropological concept of *attack*. According to this framework, negative events are thought to be due to the intervention of an exterior *attack* or a harmful intention specifically directed at the target [30]. Blaming an external source for negative events, as magical sources, can be common in traditional societies, and should not be regarded as paranoid ideation. The experience of depression and post-traumatic symptoms differ according to cultural backgrounds. Negative emotions with strong feelings of guilt can be more existent occidental Judeo Christian cultures and feelings of shame and persecution can be more current in traditional societies [31,32]. In certain cultural groups, when someone feels bad or when a person experiences successive hardships, an exterior cause is pursued (witchcraft attack, etc.). This type of cultural explanatory model [33] makes sense of the senseless [34], by making the inconceivable conceivable and facing difficult events attacking a cultural group. We can elaborate a therapeutical logic on a cultural basis when a cultural etiology is formulated. It is nonetheless difficult to elaborate on this type of etiological theory when, as for UMs, the family and its cultural group are absent. The question of what and how knowledge may be transmitted for people who leave their home at a very young age remain clinically relevant.

The proposal to use an interpreter is an ethical approach that both ensures good mutual understanding and facilitate the expression of emotions [20,35]. Moreover, the interpreter’s presence makes it possible to take distance from the classic therapist-patient dual therapeutic model, not necessarily recognized as optimal in many non-western cultures; moreover, the patient is sure to be understood and can thus be more assertive. It is important to propose an interpreter in a timely manner and to respect the patient’s refusal: using the patient’s mother tongue can remind him or her of the trauma, and the interpreter can even be identified as a member of an aggressor group.

Here, the interpreter in Lingala has been a resource as an identificatory model of migration success. We have accepted and valued the counter transferential identification movement to her father (she invests him as a symbolic father figure), in continuing, with Celine’s consent, with the same interpreter. Once, only another interpreter was available, one from Portugal and the session turned out to be completely different. The interpreter was embarrassed with the cultural representations evoked around disease in Mozambique and tend to abrade them. This highlights the importance of debriefing with interpreters [36].

For the purpose of this therapy—and being our choice in line with our specialization—we used clinical methods drawn from narrative therapy with a systemic and transcultural approach. However, we believe that taking the time to allow for strong clinical and relational bonds and a fine child psychiatry evaluation to emerge are key clinical elements, regardless of the specificity of the approach.

### 6.2. Stephano’s Reaffiliation Process… and Celine’s Reaffilation Process

An important treatment theme was to include the child in a lineage and to co-construct it.

Based on the results of a qualitative study on the outcome of 19 children conceived by rapes in the Rwandan genocide, Muhayisa (2016) describes the importance of feeling explicitly part of a—any—family [37]. The children questioned gave the impression of feeling impure and belonging to no family. Not feeling that they belonged to any lineage, even their mother’s, these children created something they could belong to by inventing a family myth. In adolescence, these young people found a sense of belonging, whatever the cost, in the face of a genealogical envelope they feel is stained or riddled with holes.

Mestre (2022) has described a feeling of debasement that persists as an indelible mark in the experience of women who have been raped [9]. Migration protects them (partially) from shame in their community of origin but exposes them to all of the other hardships linked with exile, including lack/absence, loneliness, and loss. We first worked with Celine on her own construction of her identity and her own belonging and affiliations to one or more cultural groups. We also authorized her to have ambivalent feelings toward her son, before their reaffiliation and reunion. Several protective factors have contributed favorably to the post-trauma reconstruction of her identity. The continued connection of her family group with the French State has protected her. She herself took action by fleeing. Her identity has become that of a heroic survivor and courageous adolescent in adapting herself alone to a country she did not know. She and Stephano are not isolated in France. Legal charges were instituted in her country of origin, but Celine does not know if Mozambique acted on them. The encounter with trauma specialists and social workers enabled the verbal expression and a form of recognition of the violent attack against her. The social workers have constituted an identificatory model of the “welcoming or protective French State”, as the foster mother and our clinical department. Celine was grateful for the support of the French State, but there upon worried not to be allowed to take care of her child alone as an adult.

Her reception in a foreign country and her welcome in our department and by the foster mother contributed to the form of the recognition of her need for protection. Our professional regard, reflecting benevolence rather than strangeness, foreignness or rejection, probably restored her sense that she was treated as a human being. Celine‘s mother is far, but the therapists, the foster mother, the social worker and all the actors around Celine show her that there are “symbolic mothers” around her, and that allows her to become a mother.

In this situation, factors have contributed to the re-affiliation of Stephano to Celine. He carries his mother’s last (family) name. The paternal lineage is nonexistent. Nonetheless, his name was co-constructed with the benevolent figure of the obstetrician who delivered him. An imaginary filiation is equally important in his naming [38]. Through this action, Stephano has moved from “nobody” to the person protected by the physician and the person who helped Celine to flee. She did not name him alone; he is recognized by someone from the cultural group he belongs to. Other groups in the host country show that Stephano is recognized in his role as a child to be protected: the foster mother is attached to him, and to Celine. She remains a professional responsible for receiving and sheltering this pair but also willingly accepts an identification as a second mother for both Celine and Stephano. The therapists involved also form a warm and welcoming group: the treatment team is led by a child psychiatrist, a nurse, and an intern, with the aid of an interpreter. Celine told the interpreter in Lingala: “Your intonations are the same as my father’s, they soothe me”.

Her reception in this therape”tic ’roup has allowed her to overcome her experience of injustice and the impact of microaggressions that she has experienced in school. Celine’s emotional experience was strong because of her history of trauma.

Her network of friends has also allowed Celine to be recognized and not judged. Moreover, although she had not confided her premigration history to anyone outside our treatment team, she did later confide in another young girl at the child welfare office, who had herself been raped.

In other situations, religion can also be a way of joining a community (baptism for example) and a vector of continuity.

### 6.3. Re-Creating Continuity with an Interrupted Adolescence and Hybrid Cultural Identities

Celine described her experience of mourning her family, her country, her identity as a child. Her process of adolescence was brutally interrupted by the rape and the events that followed it: exile and, motherhood. She experienced a double, even a triple, identity crisis: her status changed to that of an adolescent and to that of a mother, and in a context of trauma and exile. The re-establishment of different levels of continuity (psychic or real) was necessary to restart a process of constructing a stronger, less vulnerable identity, with a feeling of continuity as well as transformation [39]. Moro (2006) calls our attention to the particularly important indirect effects of traumatic events at two life stages: infancy and adolescence [40]. Their dependence on their relationship with their parents and parent figures means that babies and adolescents will be affected twice: first by the direct effect of the event on them, and second by the rupture of this relationship [41]. For adolescents, at a time when they are in the process of psychically separating as they grow up, when they need an effective system of projection, these events with their consequences on their bonds can make them even more vulnerable.

Babies, while they do not have representations of death, do have a theory of life that is damaged, broken, or destroyed in traumatic situations. This theory of life is linked to the concept of “fundamental trust” [42], which must therefore be repaired afterwards.

Working on family links and with families is relevant, despite the situation of UMs, so that they feel less isolated and do not become disconnected from their affiliations. Young adolescents aspire to move from childhood to adulthood, to construct their identity in a universe where secrets and silence often dominate in migration, but identity cannot be formed in a vacuum [43]. It is thus important to rebuild the bridges between before and after the elements that caused disruption.

Our work also involves improving relationships with social workers or foster parents, sometimes using limited reparenting in therapy. It is important that the child welfare bureau allowed for continuity in this family, with the same foster family and with continuity of language insofar as possible. There was no change in housing or anything else, as too often occurs for UMs. This itself can be an important aspect of treatment: UMs can have positive long-term outcomes if they are provided with appropriate health and psychosocial protection [2].

The relationship created between the foster parent and Celine allowed Celine to feel authorized to be an adolescent, with phases of regression, of conflicts possible without rupture, of the necessary reassurance about the continuity of relationships, and of affirmation by self-subjectivization. She said of herself, “sometimes I’m too much of a child, I can’t stand constraints”. Sometimes she’s too adult, her interactions marked by solemnity.

Celine seems to seek protective parental figures to identify with and with whom she can trace a sense of family continuity: the foster mother has diabetes “like my mother”. She described a good relationship with the friend of her father who helped her to leave: “he was like a father”.

One of the key points of our therapy was to not force an identity as “mother” on Celine. We supported a mixed identity, simultaneously from the cultural perspective, but also by the blending of her adolescent and adult aspects while valuing the nurturing acts accessible for her.

Celine could think of herself as a co-mother or big sister or several other kinds of parental figures for Stephano, but not yet as his mother.

Our therapy allowed her to invest in Stephano in this hybrid mode, because of the presence of the foster mother. Accordingly, our clinical priority focused on ensuring the emotional safety of Celine and her child and not on her construction of an identity as a mother at any cost, for this was inconceivable when we first met her. This “authorization” to be something else than a mother—big sister, co-mother, secret mother—provided the flexibility essential for Celine to find her way of investing herself in her child and to become an affective caregiver, little by little. Both identities could then share an emotional sphere with a process of filiation, developed in her own way, adapted and tolerable for her progression, that is, by putting Stephano psychically and emotionally at the level of her sister’s children.

### 6.4. Strengths and Limitations

#### 6.4.1. Flexibility in the Treatment Framework That Allows a Permanent Adjustment of How the Parties Speak to One Another and Respect for the Rhythm of Each Child’s Therapy

The treatment framework described in this article is itself co-constructed to be adapted to a very specific situation, which is in and of itself a strength and a limitation.

First, we focus on building a strong therapeutic alliance and sentiment of trust. In order to do so, we explicitly clarify from the start that we may talk about the trauma but we do not consider it an obligation, and that we will allow ourselves to pace the timing of our intervention according to need and to patient’s progression.

The framework can also be adapted with flexibility and handiwork to a dual progression—Celine’s and Stephano’s. We were able to separate the treatment spaces when necessary, while bringing Celine and Stephano together when a shared story might promote the reconstruction of both of their identities.

We started by exploring the impact of the migration and the levels of rupture that it has engendered. We consider that this choice made possible to avoid focusing directly on the rape and, in turn, to take into account the entirety of the vulnerable elements that have been overcome along the therapy.

We were able to start with a narrative on the mother’s side and to co-construct a new affiliation (foster family) and then to link the history of the child and his mother to a life-giving past. The story is established around those who protected them.

Moreover, the therapists valued Celine’s caring acts towards Stephano without urging into defining her role as a mother and while accepting the place that felt possible for her vis-à-vis Stephano and towards herself. Her behavior and narrative seemed to suggest she may conceive herself as an older sister, a caregiver, a mother somewhat not explicit, a co-mother, and “daughter” herself to Stephano’s co-mother.

This flexibility allowed Celine to become emotionally invested in Stephano and to make him feel/be safe with her. A narrative had to be developed from the start to name and restore little by little a shared family envelope, while protecting it from the infiltration of negative influences [44].

#### 6.4.2. What Should We Say to Children and in Front of Them?

What do we say to the children, and must we say it? The disclosure of their violent conception risks projecting fantasies on children that stigmatize them. Only a few publications have examined modalities of treatment of children conceived by rape. Some authors have shown the interest of a systemic approach, while using creative narrative supports for the mother-child pairs, such as: two genograms (one made by the mother, one by the child), which can be free and imaginary, a drawing of the family, and a symbol picturing the family line [37].

In his study of the outcome of children conceived by rape, Muhayisa also reported that their mothers, or other family member, often communicated their origins to them brutally, or thoughtlessly, in moments of anger. This revelation was then accompanied by violence, hostility, and insults to make them understand that they carried the genes of killers. It would thus be preferable for a mother who was raped to disclose this event to the child born under these circumstances in a less disruptive manner. Cotton considers that this can help bring the mother and child together, as it allows the child to understand the mother’s ambivalence which can be understood as unrelated to the child themselves or their actions [45]. Some children feel great shame, guilt for having been born. It is a relief for them to learn the truth, especially if that helps them to understand their mother’s behavior toward them.

Several authors underline that mothers who have been raped experience psychological states that they can transmit to their children without even telling them about the horror they underwent, in the form of negative transmission [46,47].

More generally, there is a debate about what can be told to children and how, regardless of the nature of the traumatic event.

Should the parents share the traumatic events they experienced to their offspring, or should they keep silent? When should they be told and how? Depending on their theoretical framework, researchers have reached different conclusions. Psychodynamic theorists have shown that a parent who has experienced a major trauma, and who has not communicated openly with his or her children may transmit the trauma to the children in the form of unconscious displaced emotion [12,41,48,49]. In situations of trauma, isolation, and ruptures, the child risks becoming the holder of the parent’s history instead of being its heir [11,50]. Whatever the form, the elements of trauma are raw and are incorporated more than introjected. Their effects can be felt for several generations, similarly to radioactive residues [11], unless they are metabolized, transformed by the psyche. We also know that there is often partial transmission in bits and pieces, which the child risks reconstructing, replaying unconsciously a traumatophilic scene later on. This calls for rethinking the intergenerational traumas not yet elaborated of close relatives.

Ngameni et al. (2022) performed a qualitative study in which they interviewed 14 mother-baby pairs; the adult mothers were all migrants who had experienced trauma, three of them rape [51].

Their findings indicate that the mothers’ choice to tell or not tell their children the story of the trauma did not depend on the type of trauma experienced. The results do, however, suggest that this choice was influenced by the intensity of post-traumatic symptoms. The level of elaboration and psychic integration of the traumatic story influence the choice of disclosure. Mothers with more severe post-traumatic symptoms were reluctant or hesitant to share their traumatic experience with their child. The recovery of these women from their trauma, through culturally appropriate therapeutic care, can effectively contribute to the choice to disclose their traumatic experiences to their children. This treatment can support them in developing open and healthy communication strategies to prevent the transmission of traumatic effects to their children.

The authors describe the importance of personalization of the traumatic experience. The mothers appropriate the traumatic story as an experience that is part of their personal life spectrum and does not directly concern their children.

Moreover, mothers describe the trap of being in-between two cultures, the one of the origin et that of the host country. Moreover, They may be ambivalent about telling their children the story of their conception or birth. This ambivalence manifests itself against the background of a cultural conflict. For some mothers, this story is not to be told as of the to the codes of their own cultural groups and that of their countries of origin. However, according to them, assimilation of the norms and codes of the host country, as well as the children’s intellectual awakening and curiosity, forces them to reveal this story to the children. The socio-educational environment of the host country forces them to a “form” of psychological transparency that contrasts with the educational practices of the country of origin.

We think that the idea of tranquility is primordial—whether she decides to tell or not tell the child. Removing the pressure to have to tell is probably itself an element of treatment. For these women, sharing their emotions and their stories with us makes their choice about telling their child legitimate and comfortable, regardless of the decision they make.

#### 6.4.3. Limits of the Intercultural

The cultural proximity of the foster family was another level of protection here. It allowed them to continue using their language and promoted cultural *metissage*, the foster mother herself being a migrant from the same country. She was both sufficiently close and sufficiently distant, because she did not belong to the exact same cultural group. We did wonder whether she might become overinvested in Stephano, which could have been confusing given his undifferentiated attachment at the beginning of care.

In other situations, it may be the reverse. We must recall that the intercultural approach is not necessarily the rule.

#### 6.4.4. Traumatic Aspects of Counter-Transference

It is important to pay attention to emotional reactions linked to the conscious or unconscious prejudices that professionals may have in any transcultural situation and act to ensure they do not impair the care relationship [52]. Nonetheless these reactions can provide informative clues about the impact of trauma in complex trauma situations [53]. Considered correctly, they can be a precious clinical material and benefit the therapeutic relationship. The clinician may be affected by the recounting of a traumatic experience, just as much as the child. Furthermore, these emotions can indicate what the child can perceive in return [12]: dissociative effects, feelings of fatigue, distress.

One of the reactions to avoid is that of vicarious trauma, which stuns, freezes thought. Working in co-therapy with several other therapists helps to avoid this risk of mental blankness. Another pitfall to avoid is misinterpreting a silent position or the failure to keep an appointment. These reactions by mothers who have been raped are often an expression of shame and psychological disorganization [9]. These women can take several years before developing the insight that allows them to articulate an inhabited narrative that combines emotion and representations.

In Celine’s situation, working on our reactions as professionals have allowed for the making of a flexible framework as well as to getting away from moralism and the injunction to be a mother at all costs often imposed on women by social and cultural norms. We believe that this was an essential element of her treatment/Celine’s therapy.

We were able to perceive Celine’s guilt for having given birth to a child conceived by rape, aggravated by her inability to represent the event. She was initially (at the early stages of therapy) unable to find way to describe the event she lived though. In the therapy sessions, we tried to work around/let emerge how Celine made sense of the violence she was submitted to and on the fact of giving birth to a child at her age (12 years old) in her specific context.

The analysis of this countertransference also makes it possible to avoid misinterpretations. For instance, what causes trauma is not necessarily what therapists with no specific training in trauma imagine. Thus, in Celine’s experience, the violence and the continuation of the pregnancy are not exclusively the major elements of her trauma, but also the absence of her family and her memories of parental conflict.

#### 6.4.5. Perspectives

The analysis of the literature shows a gap in both qualitative and prospective studies on the topic of parents’ disclosure to children about their conception as a result of rape. The goal of this article is to propose therapeutic tools as well to reflect on important questions such as: What kind of narrative is the most protective of the child’s future identity construction? How can the trauma narrative be adapted to the child’s age? How will the child face the challenges and vulnerabilities of adolescence, as a period of identity construction between affiliations and filiation, where questions of origins (knowing where I come from to know where I am and who I want to become) generally reactivate?

## 7. Conclusions

Providing care according to a culturally sensitive approach supported Celine’s recovery from trauma. In turn, it helped her to establish a healthier relationship and communication strategies with her child, thus, preventing the impacts of transgenerational trauma transmission.

Building a flexible and culturally sensitive clinical framework, as typically carried out in transcultural psychiatry, facilitated the emergence of Celine’s own narrative at her own pace.

The psychotherapy here unfolds on a dual level of progression: first, analysis of the present, then individualized and progressive care aimed at restoring a sense of security and the ability to create meaningful emotional bonds, with time allowed for the definition of the roles to take shape.

UMs are often affected by multiple traumatic events. Their resilience, psychological resources, and skills at overcoming such events may appear astonishing at times.

We consider that focusing the intervention on restoring confidence and the ability to engage in reflective and independent thinking is a primary act of care in such cases. Secure and consistent professional support helps to repair a sense of coherence where the trauma disrupted the order of things and to re-establish a sense of normalcy when one has been confronted with experiences beyond the norm.

The transcultural approach is a committed practice, politically engaged but apolitical. This clinical model is nonjudgmental and aims to restore or build meaningful bonds. We exemplify it here by illustrating a clinical frame able to reject moral injunctions on the obligation of motherhood, while prioritizing the safety of the patients and the creation of secure bonds, in a manner adapted to their experience and their progression past the trauma.

## Figures and Tables

**Table 1 healthcare-10-02036-t001:** Transcultural interview guide (From Radjack, et al., 2022 [20]).

1	Start the interview by creating the basis for an appropriate relationship explain clearly the professional’s role and treatment objectives
2	Show a willingness to mix cultural practices
3	Query the patient’s cultural identity with precision
4	Set forth the cultural explanatory models of the problem/disease
5	Explore their network of belonging and their level of isolation
6	Adapt the framework—Exit a dual relationship if a group of therapists is more culturally coherent/consistent
7	Tell the story of the migration and the premigration period
8	Identify the social stress factors and the protective factors
9	Explore the treatment itinerary and conceive a therapeutic logic
10	Take into account one’s own cultural countertransference (professional’s reactions linked to his/her own culture, in relation to the patient)

## Data Availability

Not applicable.

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
