# Peer review of "How Do We Address and Treat the Trauma of a 16-Year-Old Girl, Unaccompanied Minor, and Her Rape-Born Son? A Case Report"

_healthcare, 2022, doi:10.3390/healthcare10102036_

Round 1
Reviewer 1 Report
I admire very much the engagement visible in your paper. Difficult experiences of Celine and Stephano, giving them help, stucture and hope are very well described and are useful for every mental health proffesionals facing such patients and such problems. Focusing not only on identified patient, but noticing, observing and helping her child in flexible, not intrusive way was really inspiring, practical hint. Respect for life, dignity of the patient and letting her to decide about her own are signs of high level of proffesional skills.
Author Response
Thank you very much for your positive comments
Please find attached the new manuscript

Reviewer 2 Report
Summary:
A very interesting text. This case report deals with a topic that is not very common in the general mental health clinic, but not insignificant in the field of asylum and refugees. This case report shows how to improve the well-being of an Unaccompanied Minor (UM) mother with a child born out of rape, and how to try to prevent the impacts of transgenerational trauma transmission to his child. There is little literature on this topic, which is why this case report seems relevant to me. The article is very instructive and large in scope, well-structured and pleasant to read. It describes quite well the methodology used, but some clarifications are still needed, as mentioned later.
General concept comments:
The case report is very interesting, with many details, that allows the reader to know how the therapy has been done, and what are the key concepts to know to reproduce this kind of treatment, regarding the specificities of the patient. As the authors mentioned at the end of the text (618-622) « The recovery of these women from their trauma, through culturally appropriate therapeutic care, can effectively contribute to the choice to disclose their traumatic experiences to their children. This treatment can support them in developing open and healthy communication strategies to prevent the transmission of traumatic effects to their children”. The following idea it’s also very important (634-638) “We think that the idea of tranquility is primordial — whether she decides to tell or not tell the child. Removing the pressure to have to tell is probably itself an element of treatment. For these women, sharing their emotions and their stories with us makes their choice about telling their child legitimate and comfortable, regardless of the decision they make”. The conclusion idea is that this case report shows the interest of this kind of therapies to try to improve the well-being of the UM-mother, and to prevent the impacts of transgenerational trauma transmission”. All these ideas are relevant. I think that a synthesis of these ideas must be written before in the text, in the introduction and in the abstract. It seems to me, also necessary to mention before in the text the reactions of the therapist facing this kind of clinical situations very complicated (the counter-transference). This subject is very interesting and crucial all the time during the therapy. I would prefer to mention before this topic that you expose only at the end (648-678).
I would appreciate to better know about the relationship with the foster mother earlier in the case report. We must wait until the 521 to go deeper inside this relationship. She appears only in fragmented descriptions; but her role was major for the therapy. Not in the therapy sessions, but in the common live. I would like to better know where exactly live Celine. Did she have a room inside the foster home, or a little apartment with more privacy? The foster mother was alone, or has a family? How was the relationship with the foster father, and even with the children of the foster family? Usually, the fact to rebuild a family ambiance can help a lot to UMs, especially with this kind of psychological suffering. And what about the collaboration between the therapists and the foster mother? Did she receive some counseling about how to be with Celine and the Stepahno ?
In general, I find that there is a lack of description of the role of all the other actors of the intervention. You mentioned some information in 514-520, but not enough at my point of view. Celine had a social worker and a teacher too, and other persons who play an important role on the daily live support. Did you make some meetings with all these other actors, or only focus on the therapy sessions? This gives the impression that only the therapist did something important for Celine and Stephano, but the teachers and social workers must have played an important role as well.
Specific comments:
Following the CARE Guidelines, I have noticed some issues. The Checklist it's not completed on the "reported on line".
3b. There is not a clinical description of the main symptoms, we only find the diagnoses of complex PTSD. Nevertheless, the description of the context is extensive and clear.
3c. Not enough description of the outcomes. We find a rich description about how the therapeutical method support different kind of psychological process, but nothing about the symptoms or initial clinical issues of the patient (did the symptoms of complex PTSD improve? and which ones?).
3d. We can improve the conclusion as I mention later.
13. There is no clarification of the patient's informed consent. I think the explanations can identify the patient. It will be necessary to change the name (I imagine it is not the real one, but it should be mentioned), and also change the origins of the patient, giving other similar ones. Be aware that the language spoken is Portuguese. You can say that the patient comes from Mozambique... for example. Or remove any precision on the language spoken by the patient and the host family, saying that it was the same.
Lines:
246-248: It lacks on some explanation of the aim and some referrals for all these psychological tests “a cognitive development test (WPPSI-IV), psychomotor tests, the Bender test to assess visuo-motor functioning, R. Zazzo’s Epreuve du bestiaire,personality tests with the Rorschach and CAT tests, and the Sceno test game.” There are some explanations about the Rorschach and the Sceno Test Game later, but it would be better to introduce the referrals at the beginning and then, give some explanations as you have done for the Rorschach test, at the bottom of the page.
312-314: This sentence it’s not very clear. Celine is angry with the mother and the father? I think some commas will help to better structure the sentence.
339-346: This paragraph is very rich with, at my point of view, key concepts that the reader needs to better know. If the reader is an expert on the field, he will quickly understand the concepts, and referrals, but for not experts it can be difficult, and lost the opportunity to introduce master ideas about the transcultural approach. I would appreciate that the authors explain that the “attack” it’s a sort of magical explanation of an event, and it would be great to explain what the Kleinman’s concept of cultural explanatory model (21) is and why it’s so important to understand that. Finally, I would appreciate also to introduce a little bit more the referral of Zempléni (22).
386: A mistake “that” must be “that”.
395: A more accurate explanation of “cultural metissage” is needed to better explain to the reader what means and how to manage. A referral it will be great too. I can bring you some if needed.
430: Another time, it seems important to me to better describe “to mix cultural practices” (see cultural metissage). A referral is necessary at my point of view.
438: The interpreter can be identified as a problem, but also as a resource as well (an identificatory model of migration success), or become as a screen projection of the transferential movement of the patient. At your point of view, which analyses do you do about this relationship, between Celine and the interpreter, and how you managed this relationship in the sessions? Did you work always with the same interpreter? The collaboration was easy, or you needed to work on it with the interpreter to improve the co-working? You mentioned something very interesting, at my point of view, in the 479-480, a transference movement from Celine to the interpreter. What did you do with this movement?
439-489: I think that it’s necessary to review the title and the main idea of this point. I think that in this part you talk about the reaffiliation process of Stephano, but of Celine as well. To become a mother, Celine needs to feel that she has a mother who shows her how to become a mother. Her mother is far, but I think that the therapist, the foster mother, the social worker and all the actors around Celine, shows her that there are "Symbolic mothers" around her, and that allows her to become a mother.
456-457: At my point of view, in this paragraph you can introduce a larger explanation of the role of the social workers as an identificatory model of the “welcoming or protective French State”. As the foster mother, as you as therapist and your clinical department. How Celine perceives, conceives, the support of the French State? Did she explain something about?
479-480: You mentioned something very interesting, at my point of view, a transferential movement from Celine to the interpreter. We can interpret it as “she invests him as a symbolic father figure”. What did you do with this movement?
554: mistake. A “-“ is before “The framework…”
602-604: I think that we can also add the referral of Serge Tisseron “Secrets de famille”. He has developed a lot these ideas of trauma transmission.
Thank you very much for this case report !
Reviewer 3 Report
The article has great practical value.The description of mother and child psychotherapy in a situation of trauma and cultural differences is exemplary and constitutes valuable material for psychologists and psychiatrists.The article is a description of 2 years of work of the therapeutic team. It may constitute a standard of conduct for other therapeutic teams.The conclusions are in line. And the references are appropriate. What procedure of psychotherapy should be used in the case of an immigrant underage girl who has experienced rape and is the mother of a child?
Author Response
Thank you very much for your positive comments. For the purpose of this therapy – and being our choice in line with our specialization - we used clinical methods drawn from narrative therapy with a systemic and transcultural approach. However, we believe that taking the time to allow strong clinical and relational bonds and a fine child psychiatry evaluation to emerge are key clinical elements, regardless of the specificity of the approach.We added this item in the discussion (line 530)

Reviewer 4 Report
Dear Authors
Very interesting presentation of the effects and the treatment of psychological trauma. However, you should pay attention in some points:
Unaccompanied minors are a vulnerable population at risk of post-traumatic
stress symptoms, what is special about this article
Make the purpose of the study clear
At the beginning of the case report, describe the main symptoms and then continue with the analysis of the case and the history.
Where did you get permission to publish the article?
Was the girl informed and consented or someone else?
Where was the guide in table 1 taken from? Cite by reference
Also, apart from the team of psychotherapists, which health professionals participated in the treatment?
Table 1 also, may have footnotes
The section Strengths and limitations is quite long
References must be formatted according to the journal's requirements
Regards
